# RSSI Fingerprint Height Based Empirical Model Prediction for Smart Indoor Localization

**DOI:** 10.3390/s22239054

**Published:** 2022-11-22

**Authors:** Wilford Arigye, Qiaolin Pu, Mu Zhou, Waqas Khalid, Muhammad Junaid Tahir

**Affiliations:** 1School of Communication and Information Engineering, Chongqing University of Posts and Telecommunications, Chongqing 400065, China; 2Engineering Research Center of Mobile Communications, Ministry of Education, Chongqing 400065, China

**Keywords:** path–loss modeling, fingerprinting, WLAN, indoor localization

## Abstract

Smart indoor living advances in the recent decade, such as home indoor localization and positioning, has seen a significant need for low-cost localization systems based on freely available resources such as Received Signal Strength Indicator by the dense deployment of Wireless Local Area Networks (WLAN). The off-the-shelf user equipment (UE’s) available at an affordable price across the globe are well equipped with the functionality to scan the radio access network for hearable single strength; in complex indoor environments, multiple signals can be received at a particular reference point with no consideration of the height of the transmitter and possible broadcasting coverage. Most effective fingerprinting algorithm solutions require specialized labor, are time-consuming to carry out site surveys, training of the data, big data analysis, and in most cases, additional hardware requirements relatively increase energy consumption and cost, not forgetting that in case of changes in the indoor environment will highly affect the fingerprint due to interferences. This paper experimentally evaluates and proposes a novel technique for Received Signal Indicator (RSSI) distance prediction, leveraging transceiver height, and Fresnel ranging in a complex indoor environment to better suit the path loss of RSSI at a particular Reference Point (RP) and time, which further contributes greatly to indoor localization. The experimentation in different complex indoor environments of the corridor and office lab during work hours to ascertain real-life and time feasibility shows that the technique’s accuracy is greatly improved in the office room and the corridor, achieving lower average prediction errors at low-cost than the comparison prediction algorithms. Compared with the conventional prediction techniques, for example, with Access Point 1 (AP1), the proposed Height Dependence Path–Loss (HEM) model at 0 dBm error attains a confidence probability of 10.98%, higher than the 2.65% for the distance dependence of Path–Loss New Empirical Model (NEM), 4.2% for the Multi-Wall dependence on Path-Loss (MWM) model, and 0% for the Conventional one-slope Path-Loss (OSM) model, respectively. Online localization, amongst the hearable APs, it is seen the proposed HEM fingerprint localization based on the proposed HEM prediction model attains a confidence probability of 31% at 3 m, 55% at 6 m, 78% at 9 m, outperforming the NEM with 26%, 43%, 62%, 62%, the MWM with 23%, 43%, 66%, respectively. The robustness of the HEM fingerprint using diverse predicted test samples by the NEM and MWM models indicates better localization of 13% than comparison fingerprints.

## 1. Introduction

Wireless communication in recent years has seen a gradual growth of reliance on services-based localization, justified by Artificial Intelligence optimization algorithms [1,2], opening up huge prospects. Even though outdoor Global Navigation Satellite Systems (GNNS) such as the Global Positioning System (GPS) [3,4], BeiDou (BDS) [5], Global’nalya Navigastsionnaya Sputnikovaya Sistema (GLONASS) [6], have been applied in outdoors scenarios, they render inadequate assistance within a complex in indoor environments [7], attributed to the complexity of indoor environment [8], due to multipath interference, shadowing, dynamic environmental changes due to observed variation effects of different materials inside the building on the dissemination of signal and the need for higher accuracy at a lower cost and no extra hardware support [9]. The widespread use of sensors, smartphones, and the mobile Internet has allowed precise mobile positioning [10]. A milestone in the realization of the Internet of Things (IoT) has seen a vast increase in the deployment and application of smart indoor environments that dramatically drive the Location-Based Services (LBS) concept, ranging from increased social networking to health applications such as healthy aging monitoring, personal tracking, enhanced 911 (E911) emergency response indoor route analysis, inventory control, wall through intruder detection, drone tracking and robotics, peer file sharing and printing applications, in-airport passage assisted navigation [11] and several other indoor location-aware applications proportionally contributed to the research objectives and motivation, without forgetting location side-information that can enable environment aware communication network design, operations, and optimization in high radio spectrum of the 5G new radio and the 6G networks [12,13,14,15]. The key enablers can be attributed to new frequencies, the development of Artificial Intelligence (AI), and Machine Learning (ML) techniques over the years.

Research has considered specific sensors and emitters installed inside a target indoor space to localize user equipment and objects. It is not scalable and includes a specific overhead cost to install and maintain the extra infrastructure. Different technologies such as ultra-wideband [16], Bluetooth [17,18], Radio Frequency Identification (RFID) [19], Micro-electro-mechanical (MEMS) [20], magnetic field [21], ultrasonic [22], computer vision [23], infrared signal [24] and other’s utilize existing infrastructure such as the WLAN [25], that considers wireless RSSI to localize the user equipment, RSSI values readings are widely and freely available in most mobile devices of this IoT era, not forgetting the availability in the most popular operation system such as the android and the Microsoft windows [26,27]. This approach normally needs comprehensive pre-surveying and training efforts to establish radio frequency (RF) characteristics of the complex indoor environment.

Measuring the distance between the unknown node and beacons is an essential part of the positioning process within indoor environments. Most of the existing UE localization algorithms used nowadays can be divided into two categories depending on whether distance measurements are required or not. One of these categories is the range-free measurement localization algorithm, and the other is the range-based measurement localization algorithm [28]. The distance measurement algorithm calculates the distance between the known beacon node and the unknown node connected to it, utilizing their communication link parameters. The main categories of distance measurement algorithms are the Angle of Arrival (AOA) based algorithm [29,30], Time of Arrival (TOA) based algorithm, and Time-Difference of Arrival (TDOA) based algorithm [31,32,33]. The TOA and TDOA require synchronization and accurate timing for components, thus, increasing the complexity of the system, and AOA for localization will require realistic antenna arrays that require high energy consumption, cost, time, additional hardware, and the RSSI based-algorithm [34,35,36,37,38,39]. Keeping in mind that each of these considerably increases the cost of the positioning system, thus, the proposed HEM indoor technique is based on RSSI prediction and distance measurement.

Indoor radio propagation environment has in the past and currently been perceived as a random component of advancing wireless communication systems, which has led researchers to introduce Intelligent Reflective Surfaces (IRSSI) as a promising solution to control scattering, reflection and refraction by allowing dynamic shaping and control of the electromagnetic waves responses of the environmental objects through the phase, amplitude, frequency and polarization parameters [40] IRSSI could enable tracking/surveillance application in Non-Line of Sight (NLOS) communications and autonomous localization. Amongst the breakthroughs, utilization of the RSSI-based localization approach, such as fingerprinting, has been adopted for location estimation and wireless coverage estimation methods [41,42,43,44,45]. In [46], a fingerprint-based positioning algorithm is proposed by collecting RSSI samples into the fingerprint database. In [47], the authors have executed and analyzed several positioning algorithms such as centroid localization, proximity localization based on RSSI, fingerprinting, and trilateration localization, conclusive that the fingerprinting positioning algorithm is the most fitting one. Fingerprint positioning methodologies require a large amount of a priori information support, which adds a high–cost issue. Moreover, suffering from multipath effects and electromagnetic interference due to different types of materials have a significant impact on wave propagation and as a function of the frequency in complex indoor environments. Such an approach considers two phases, the offline phase, and the online phase. During the off-line calibration phase, the target indoor environment is calibrated with the help of a pre-site survey to predetermine grid reference points at which time-stamped sampled signal values as fingerprints from various transceivers are recorded and stored in a database commonly known as the “radio map.” The online phase, commonly known as the “Localization” phase, will deploy various algorithmic approaches to effectively match the stored radio map values to find the best match, whose location is presented as the localization result. This approach’s drawback relates to the size of the database when handling a large number of observations and the time spent. The mode for the indoor propagation channel has a significant impact on RSSI value, which shows variability with changing locations [48]. This variability is based on many effects of the separation distances, the geometrical, different materials used, and the movement of the objects. Additionally, multipath and shadow fading have a great effect on RSSI values [49]. In contrast, RSSI-based location estimation is significantly affected by the position of the AP and the position of received points. Several researchers have proposed AP location estimation using neural networks [50] as well as studying the effect of transmitter placement in the wireless sensor network and Line of Sight (LOS) investigation of different AP heights on RSSI measurement variation as in [51]. These proposed methods have not leveraged the indoor Fresnel ranging coverage of interest based on transceiver height in LOS and NLOS in a multi-wall indoor environment to improve the path–loss RSSI prediction model that, in turn, reduces the cost of the RSSI fingerprinting.

This paper proposes a novel indoor positioning technique based on a new RSSI distance prediction HEM model that leverages the transceiver height, the signal wavelength, and its assumed Fresnel coverage of interest on the target floor. The proposed technique based on a novel method for RSSI prediction improves both the estimation and localization accuracy for complex indoor environments at a low cost of fingerprinting since no additional hardware is required. Our proposed method is experimentally tested and verified in an indoor laboratory environment and a corridor using six WLAN APs. In addition, real RSSI data measurement was collected using our developed android application (APP), and data were processed for conventional assessment of the performance of the proposed methodology. In this paper, our contributions may be summarized in the following parts:Novel transceiver height and signal wavelength dependence on RSSI path–loss model prediction are proposed. A large number of RSSI samples are predicted, and each RSSI sample is formulated into fingerprints quickly. Reducing the complexity and technical know-how required for offline conventional fingerprinting.We propose an indoor radio fingerprint-based approach to the calculation of the value of *k* as in the number of nearest neighbors for the K-Nearest Neighbors algorithm leveraging the surface area of the target space to the number of sampling points.

The organization of this paper is as follows. Section 2 presents related works on the propagation models for fingerprint architectures. In Section 3, the presentation of the proposed transceiver height dependence model (HEM), data collection, Wi-Fi signal acquisition setup, model prediction setup, and prediction accuracy discussion. Section 4 presents the Localization evaluation of the proposed fingerprint toward existing approaches. The final remarks are presented in Section 5.

## 2. Related Work

RSSI localization techniques measure signal strength from a UE to several hearable APs and then combine this information with a propagation model to determine the distance between the client device and the access points. RSSI is considered the simplest approach for ranging since almost no additional cost is required for collecting the RSSI data, which is provided by most systems enabled with wireless Network Interface Cards (NICs) [52]. In basics, it is the measurement of received radio signal power as the ratio of measured power decibels (dB) to one mill watt (mW). However, it is also a less accurate way due to complicated environmental impacts on the Radio Frequency (RF) signal propagation with multipath fading. Therefore, the RSSI radio map, which is used to translate the signal strength into distance estimation, should be calibrated for every single antenna to achieve better results. The initial solution is to measure the RSSI values at all possible RP’s with predefined density and renew the mapping periodically, which is not practical to maintain such a system in ever-changing complex environments.

### 2.1. Fingerprint Architecture

RSSI fingerprinting architecture, widely viewed as the improved version of the RSSI technique, considers the preliminary offline calibration phase and the secondary localization online phase. The underlying difference is that its focuses on pre-recording the signal strength values from detectable RF transceivers such as Base Stations (BTS) (for GSM signals), AP (for Wi-Fi signals), and Frequency Modulation (for FM). The pre-recorded signal strength values information paired with the client UE’s known Cartesian coordinates of calibrated RP’s are stored in a database structure [53], with dimensionality directly proportional to the detectable number of transceivers at a particular point. During the secondary localization online phase, the current detectable transceiver signal strength vector at an unknown location is correlated to the stored fingerprint vectors in the database structure to find the closest match, then returns it’s known and tagged Cartesian coordinates as the estimated UE location subject to the localization algorithm criterion. In literature, these algorithms deploy various approaches to estimate the locality. Among these, we find distance-based (deterministic) approaches, neural networks approach, probabilistic approaches, etc. [54,55,56,57]. Amongst indoor probabilistic approaches, a Bayesian network that leverages fingerprinting to model a 3D Bayesian model (3D-BGM) was presented by bounding the predicted UE’s location by the testbed dimension, resulting in high localization accuracy with the use of a small-size radio map [58] than the author’s introduced 2D Bayesian network [59]. Fingerprinting is advantageous since it does not require the UE’s LOS to RF transceivers to estimate the unknown location; however, it is time-consuming in cases of larger target environments. Figure 1 illustrates the RF system architecture of a fingerprint-based localization system. This system comprises the infrastructure module, the offline RF sampling, the database construction module, and the online localization phase module.

Fingerprinting approaches rely on big data acquisition, which in turn leads to the considerable time and effort required to build the preliminary offline database, which highly relays on the indoor dimensionality in terms of size to calibrate RP’s and number of location candidates [60], with each RP fingerprint, is an average of total RSSI samples received within a specific time window. If the indoor localization target floor space is huge in dimensionality, then the cost of the survey increases dramatically, not forgetting the effect of environmental changes. This kind of approach encounters disadvantages not limited to the changes in the environment, such as demarcating the indoor space, which may change the fingerprint correspondence to each location, thus, the requirement to update the fingerprint database.

### 2.2. Path–Loss Prediction Methods

Building a radio fingerprint database has proven to be intense, involving but not limited to the extensive calibration phase and expertise. Several approaches have been thought about, such as interest in predicting the path–loss of the indoor WLAN-based Real-time Locating Systems (RTLSs), though an extremely challenging task because there are too many indoor-specific parameters [61]. On the other hand, many studies have attempted to approximately predict the RSSI fingerprint instead of discovering an omnipotent model that accounts for every parameter. One category is very accurate and site-specific, and it can predict wide-band parameters (Deterministic approach). However, it has a higher computational complexity and requires pre-processing and simplification procedures. The second category, although not as accurate as the deterministic model, can be easily computed because of its simple design and its smaller size input (Empirical approach). Based on these observations, we chose to use the empirical model for our propagation model while paying attention to three well-known empirical models: the one-slope model (OSM), the multi-wall model (MWM) [62], and the new empirical model (NEM) [63], respectively.

#### 2.2.1. Conventional One-Slope Path-Loss (OSM)

RSSI ranging methods predict the distance between the receiving signal node and the transmitting signal node by measuring the received signal strength since the propagation loss affects the transmitted wireless signal. The model of signal propagation follows the log-distance distribution as shown in Equation (1).
(1)$$PL\left(d\right)\left[dB\right]=PL\left(d_{0}\right)\left[dB\right]+10nlog\left(\frac{d}{d_{0}}\right)$$
where $PL\left(d_{0}\right)$ is the RSSI reference path–loss at separation distance $d_{0}$ from the transceiver; PL(d) is the RSSI at the receiving node at separation distance d from the transceiver and n specifies the path–loss propagation exponent that takes diverse behavior for a particular type of building. In complex indoor environments, $d_{0}$ is assumed to be 1 m, and n may considerably vary from approximately 2 for the LOS paths up to 6.5 for the highly abstracted path.

#### 2.2.2. Multi-Wall Dependence on Path–Loss (MWM)

A complex indoor environment complies with multiple walls and floors of different build materials that contribute highly towards the decomposing signal; as the signal traverses each wall or floor, it imposes a loss (dBm), thus, lower signal power at the point of the receiving UE. Signal prediction with the wall and floor dependence on path–loss can be defined as in Equation (2).
(2)$$PL\left(d\right)\left[dB\right]=PL{\left(\frac{d}{d_{0}}\right)}^{n}+\sum_{p=1}^{P}WAF\left(p\right)+\sum_{q=1}^{Q}FAF\left(q\right)$$
where P and Q are the total number of walls and the total number of floors between the transmitter and receiver, respectively. The empirical parameters WAF(p) and FAF(q) are termed the pth wall attenuation factor and the qth floor attenuation factor, respectively. In general, the values of these parameters are determined by the measurement data of the target building. However, several studies investigate the values of the attenuation factors of several materials in different frequency bands [53,64].

#### 2.2.3. Distance Dependence of Path–Loss Exponent (NEM)

Although the MWM performs well in certain circumstances, it does not include such propagation effects as the distance dependence of the path–loss exponent, the angle dependences of the WAF(p) and FAF(q) or the refraction and diffraction. As a result, the MWM’s prediction accuracy may be poor in certain parts of the building, especially at large distances from the transceiver. The distance dependence of the path–loss exponent can be defined as in Equation (3) [63].
(3)$$\begin{array}{l} PL\left(d\right)\left[dB\right]=10\log PL{\left(\frac{d}{d_{0}}\right)}^{n1}U\left(d_{bp}-d\right)+10\left[\log{\left(\frac{d_{bp}}{d_{0}}\right)}^{n1}+\log{\left(\frac{d}{d_{bp}}\right)}^{n2}\right]U\left(d-d_{bp}\right)\\ +\sum_{p=1}^{P}WAF\left(p\right)+\sum_{q=1}^{Q}FAF\left(q\right) \end{array}$$
where $d_{bp}$ is the distance of the breakpoint from the transceiver, $n_{1}$ and $n_{2}$ are the path–loss exponents on either side of the breakpoint, and U(.) is the unit step function defined as
(4)$$U\left(d\right)=\{\begin{array}{c} 0,\;\;d<0\\ 1,\;\;d\geq 0 \end{array}$$

#### 2.2.4. Angle Dependence of Attenuation Factors

When electromagnetic radiation is incident on a wall or floor, obliquely less signal power will be transmitted through the wall than would occur at normal incidence. To try and capture this effect in the model, we incorporate the angle of incidence effect into the WAF(p) or the FAF(q). At grazing incidence, we assume that transmission is zero, while at normal incidence, we take transmission as the value of WAF(p) or FAF(q). At angles between grazing and normal incidence, we calculate these values using a cosine function $WAF\left(p\right)/\left(\cos\theta_{p}\right)$ where the WAF(p)[dB] is taken as the attenuation factor at normal incidence and $\theta_{p}$ is the angle of incidence at the pth wall. From studies results, it can be observed that it performs well, especially when compared to what would be obtained if no variation of WAF(p) with incident angle is allowed.
(5)$$\begin{array}{l} PL_{∠}\left(d\right)\left[dB\right]=10\log{\left(\frac{d}{d_{0}}\right)}^{n1}U\left(d_{bp}-d\right)+10\left[\log{\left(\frac{d_{bp}}{d_{0}}\right)}^{n1}+\log{\left(\frac{d}{d_{bp}}\right)}^{n2}\right]U\left(d-d_{bp}\right)\\ +\sum_{p=1}^{P}\frac{WAF\left(p\right)}{\cos\theta_{p}}+\sum_{q=1}^{Q}\frac{FAF\left(q\right)}{\cos\theta_{q}} \end{array}$$
where WAF(p) and FAF(q) are the values of the attenuation factors at normal incidence, and the $\theta_{p}$ and $\theta_{q}$ are the angles, respectively, between the pth wall, qth floor, and straight-line path joining the transmitter to the receiver. The subscript ∠ is used PL(d) to indicate that it is the path loss when the angle of incidence to the wall and the distance dependence of the propagation are taken into account [63].

## 3. Proposed Transceiver Height Dependence Model (HEM)

Indoor environments have continued to be more complex and challenging, attributed to interior design finishes, space demarcating, and occupancy. Thus offline RSSI estimation from the transceivers using models is affected by several factors not limited to the multi-path effects (reflection, refraction, and absorption). Considering online localization techniques based on RSSI distance prediction using the RSSI log-distance distribution model is challenging because severe RSSI fluctuation occurs, especially in a complex indoor environment. In our study, we propose a new RSSI prediction technique that leverages the transceiver height into the formulation of the prediction model, which in turn minimizes the prediction error as well as localization error. Determination of the electromagnetic field region around a vertically installed dipole antenna, regions such as the reactive near-field region, radiating near-field region, also known as the Fresnel region, and the far field region (Fraunhofer) is a more challenging task within indoor space than related studies for outdoor scenarios. For example, considering a vertical monopole in a complex indoor space, such as an AP transceiver working at a height h(meters) of 2.4 GHz, we propose to define the breakpoint distance dependent leveraging the transceiver height and wavelength of the signal as defined in Equation (6).
(6)$$d_{bp}\approx 0.62\sqrt{\frac{h^{3}}{\lambda }}$$
(7)$$\lambda =\frac{c_{0}}{f}$$
where $d_{bp}$ is the proposed breakpoint distance ranging region, h is the maximum dimension of the antenna, λ is the wavelength of the transmitted signal by the antenna, which is further derived as a fraction of the speed of light $c_{0}$ to the frequency of the transmitted signal f as defined in Equation (7). Considering Equation (6) into Equation (7) leads to the newly proposed novel path loss is denoted as:(8)$$\begin{array}{l} PL_{h}\left(d\right)\left[dB\right]=10\log{\left(\frac{d}{d_{0}}\right)}^{n1}U\left(0.62\sqrt{h^{3}/\frac{c_{0}}{f}}-d\right)\\ +10\left[\log{\left(\frac{0.62\sqrt{h^{3}/\frac{c_{0}}{f}}}{d_{0}}\right)}^{n1}+\log{\left(\frac{d}{0.62\sqrt{h^{3}/\frac{c_{0}}{f}}}\right)}^{n2}\right]\\ {}*U\left(d-0.62\sqrt{h^{3}/\frac{c_{0}}{f}}\right)+\sum_{p=1}^{P}\frac{WAF\left(p\right)}{\cos\theta_{p}} \end{array}$$
(9)$$U\left(d\right)=\{\begin{array}{c} 0,\;\;d<0\\ 1,\;\;d\geq 0 \end{array}$$

### 3.1. RSSI Data Collection

All the experimental tests are performed at a faculty-building floor of the School of communication and information engineering, Chongqing University of Posts and Telecommunications (CQUPT). A cubic meters floor of the west wing, as shown in Figure 2, was considered for the experimental data fingerprinting benchmark to verify the extent to which our newly proposed HEM model, as defined in Equation (8), has applicable results in the current drive for smart IoT location-based communication. The test bed of 56.93 m × 20.08 m is comprised of rooms, corridors, offices, and washrooms with diverse floor and ceiling finishes, a typical indoor environment, during the daytime to idealize the impact of occupancy in the environment. Starting from the left side of the test bed as the origin marked in red color (0,0), we carried out the tiresome calibration of the site to obtain 88 RP locations spaced at 0.6 m apart in the two corridor areas, as areas 1 and area 2 before the lifts, respectively, and in the research lab as area 3, thus, dividing the space into three areas. As shown in Figure 2, the black circle and the red triangle represents the reference points (RP’s), and the test points (TP’s), respectively. In general, they are comprised of 48 RPs and 11 TPs in area 1, 20 RPs and 6 TPs in area 2, and 20 RPs and 8 TP in area 3, with a total of 6 D-Link DAP-2310 Aps with known Mac addresses as shown in Table 1, arranged in the target environment at a height (h) of 1.89 m. A realistic environment was considered, with occupants on the floor, some walking randomly and others while working on their tables.

### 3.2. Wi-Fi Signal Acquisition

The actual sampling acquisition of RSSIs in the experimental environment using our developed Wi-Fi signal data acquisition software comprises the process of collecting RSSI samples of each DAP-2310 AP using the Samsung S7568 mobile phone terminal setup at the level of the pedestrian arm length height as shown in Figure 3 and Figure 4, respectively.

Our developed Wi-Fi signal data acquisition software interface is simple and user-friendly, composed of the RP name and the time interval at which we sample the RSSI (in dBm) in a convenient way and preprocess the raw data, as shown in Figure 4a. On initiating the process, as illustrated in Figure 4b for a predefined time stamp, it records the sampling time interval, the RSSI recording, followed by the MAC address of the transceiver AP as shown in Table 2. The Samsung S7568 mobile phone has a Wi-Fi signal acquisition frequency of 1 Hz, thus, enabling us to first collect RSSI samples from 6 APs at each respective RP for a duration interval of 60 s (that is, the RSSI sample of an AP at each RP contains 60 RSSI values). Similarly, in the same faith, at each TP, the same Samsung S7568 smartphone is used to collect RSSI data samples from 6 APs for 20 s (that is, the RSSI sample from a certain AP at each test point contains 20 RSSI values). The data is then saved in the Security Digital card (SD) as a txt. format to be processed using MATLAB R2013a as a programming tool to construct the location fingerprint Database to achieve the proposed methodology of this article.

Letting N be the number of RPs and L the total number of APs deployed in the signal coverage target floor. We denote the RSSI value from AP l at RP i as $f_{i}^{l}\left(d\mathrm{Bm}\right)$. We sample multiple random fingerprint signals at each predefined RP, then average the signal values to find the mean RSSI ${\bar{f}}_{i}^{l}\left(d\mathrm{Bm}\right)$ at each RP i from AP l denoted as
(10)$${\bar{f}}_{i}^{l}=\frac{1}{s_{i}^{l}}\sum_{s=1}^{s_{i}^{l}}f_{i}^{l}\left(s\right),\;\;\;\;i=1,\ldots ,N.\;\;\;\;\;\;\;\;l=1,\ldots ,L$$
where $f_{i}^{l}\left(s\right)$ is the sth RSSI sample (in dBm) at RP i from AP l, and $S_{i}^{l}$ is the total number of RSSI samples collected within the predefined time stamp. Then the fingerprint at RP i is defined as
(11)$$F_{i}=\left[{\bar{f}}_{i}^{1},{\bar{f}}_{i}^{2},\ldots ,{\bar{f}}_{i}^{L}\right]$$

Forming an interactive radio map database matrix as
(12)$$\Psi =\left(\begin{array}{ccc} {\bar{f}}_{1}^{1} & \cdots & {\bar{f}}_{1}^{L}\\ \vdots & \ddots & \vdots \\ {\bar{f}}_{N}^{1} & \cdots & {\bar{f}}_{N}^{L} \end{array}\right)$$
where ${\bar{f}}_{1}^{1}=0$ when the lth AP is not detected, however, in some cases, we convert it to −90 dBm as the signal noise floor within an indoor environment.

Online localization exploits the pre-constructed database to determine the current location $\hat{\ell }$ given the σth test RSSI sequence denoted as
(13)$$\vartheta_{i}^{j}\left(\sigma \right),\sigma =1,\ldots ,{\hat{b}}_{i}^{j},\;\;\;\;\;{\hat{b}}_{i}^{j}>1$$
where ${\hat{b}}_{i}^{j}$ is the total number of received test RSSI sequences from jth AP at the ith unknown test RP l. Taking into consideration each test fingerprint as $\vartheta =\left[{\bar{\vartheta }}_{i}^{1},{\bar{\vartheta }}_{i}^{2},\ldots ,{\bar{\vartheta }}_{i}^{Y}\right]$ we define the test localization radio map as $\Lambda =\vartheta_{DY}$ by
(14)$$\Lambda =\left[\begin{array}{ccc} {\bar{\vartheta }}_{1}^{1} & \cdots & {\bar{\vartheta }}_{1}^{Y}\\ \vdots & \ddots & \vdots \\ {\bar{\vartheta }}_{D}^{1} & \cdots & {\bar{\vartheta }}_{D}^{Y} \end{array}\right]$$
where
(15)$${\bar{\vartheta }}_{i}^{j}=\frac{1}{{\hat{b}}_{i}^{j}}\left(\sum_{\sigma =1}^{{\hat{b}}_{i}^{j}}\vartheta_{i}^{j}\left(\sigma \right)\right),\;\;\;\;\;\;\;i=1,\ldots ,D.\;\;j=1,2,\ldots ,Y$$${\bar{\vartheta }}_{i}^{j}$ is the averaged recorded test RSSI sequence from the jth AP at the ith unknown test RP, D is the total number of query test reference points, Y is the total number of detectable AP’s. The squared Euclidian distance $d_{i}^{2}$ between the fingerprints ${\bar{f}}_{i}^{1}$ and the observed fingerprint ${\bar{\vartheta }}_{j}$ is given by
(16)$$d_{i}^{2}={\sum_{j=1}^{Y}\left({\bar{f}}_{i}^{1}-{\bar{\vartheta }}_{j}\right)}^{2},\;\;\;\;\;\;i=1,\ldots ,D$$

Considering the random signal level mean during the data processing, we differentiate the RSSI values within an indoor environment using the mW instead of dBm, i.e.,
(17)$${{\bar{f}}_{i}^{l}|}_{mW}=10^{\left({{\bar{f}}_{i}^{l}|}_{dBm}\right)/10}$$
which transforms RSSIs from smartphones to values for better signal differentiation. Correspondingly we also transform $t^{l}’s$ RSSI values in ϑ from dBm into mW.

### 3.3. The RSSI Model Prediction Setup

Primarily prediction parameters from other studies could be obtained from consideration of theoretical definitions or electromagnetic simulations or performing limited propagation experiments in the building, thus, requiring site-specific information to be obtained. Increasing the need for expertise and cost of fingerprinting in the vast emerging complex indoor environments for the IoT APP. In our one-floor evaluation approach, we eliminated the FAF parameters and defined the WAF parameters based on wall type properties in previous studies, as tabled in Table 3, to construct an RP to WAF sequence at the ith RP given by
(18)$$WAF_{i}\left(p\right),i=1,\ldots ,N,\;\;\;\;\;p=1,\ldots ,P$$
where P is the total number of walls between the transceiver and the receiver on the target floor and N is the total number of RPs, respectively.

Taking into consideration each wall attenuation factor from pth wall at the ith RP, $WAF_{i}=\left[\Psi_{i}\left(1\right),\Psi_{i}\left(2\right),\ldots ,\Psi_{i}\left(P\right)\right]$, we define the wall attenuation map as $WAF_{NP}$ by
(19)$$WAF_{NP}=\left[\begin{array}{ccc} \Psi_{1}\left(1\right) & \cdots & \Psi_{1}\left(P\right)\\ \vdots & \ddots & \vdots \\ \Psi_{N}\left(1\right) & \cdots & \Psi_{N}\left(P\right) \end{array}\right]$$

The pre-offline association of the wall intercept with the 2-Dimensional vector segment between the RP and the AP Cartesian coordinate location is formulated so as to simplify the all attenuation map to only intercepts. Let line segment one S1 be defined as the segment between the ith RP and the lth AP, that is $S1=\left[X_{1}\left(l\right)\;\;\;\;Y_{1}\left(l\right)\;;\;\;X_{2}\left(i\right)\;\;\;\;Y_{2}\left(i\right)\right]$, and line segment two S2 be a segment between the 2-Dimensional end of the pth wall, that is $S2=\left[X_{3}\left(p\right)\;\;\;\;Y_{3}\left(p\right)\;;\;\;{\hat{X}}_{4}\left(p\right)\;\;\;\;{\hat{Y}}_{4}\left(p\right)\right]$, respectively, to form a matrix
(20)$$\left[\begin{array}{c} X\\ Y \end{array}\right]=\left[\begin{array}{c} X_{1}\left(l\right)\;\;\;\;X_{2}\left(i\right)\;\;\;\;X_{3}\left(p\right)\;\;\;\;X_{4}\left(p\right)\;\\ Y_{1}\left(l\right)\;\;\;\;Y_{2}\left(i\right)\;\;\;\;\;Y_{3}\left(p\right)\;\;\;\;\;Y_{4}\left(p\right) \end{array}\right]$$

We defined the intersection of the lines by solving determinates of the matrix as in Equations (21) and (22).
(21)$$dt1=\left|\begin{array}{ccc} 1 & 1 & 1\\ X_{1}\left(l\right) & X_{2}\left(i\right) & X_{3}\left(p\right)\\ Y_{1}\left(l\right) & Y_{2}\left(i\right) & Y_{3}\left(p\right) \end{array}\right|*\;\;\;\left|\begin{array}{ccc} 1 & 1 & 1\\ X_{1}\left(l\right) & X_{2}\left(i\right) & X_{4}\left(p\right)\\ Y_{1}\left(l\right) & Y_{2}\left(i\right) & Y_{4}\left(p\right) \end{array}\right|$$
(22)$$dt2=\left|\begin{array}{ccc} 1 & 1 & 1\\ X_{1}\left(l\right) & X_{3}\left(i\right) & X_{4}\left(p\right)\\ Y_{1}\left(l\right) & Y_{3}\left(i\right) & Y_{4}\left(p\right) \end{array}\right|\;*\;\left|\begin{array}{ccc} 1 & 1 & 1\\ X_{2}\left(l\right) & X_{3}\left(i\right) & X_{4}\left(p\right)\\ Y_{2}\left(l\right) & Y_{3}\left(i\right) & Y_{4}\left(p\right) \end{array}\right|$$
where the intersect relation Inter(i,p) by the ith RP and the pth wall is given by
(23)$$Inter\left(i,p\right)=\{\begin{array}{c} 1,\;\;\;\;\;\;\;\;\;\;(\;dt1\leq 0\&\&\;dt2\leq 0)\\ 0,\;\;\;\;\;\;\;\;\;(\;dt1>0\&\&\;dt2>0) \end{array}$$

Having obtained a matrix that corresponds to the intersect relation, the total contribution of the wall attenuation factor WAF(p) and RP is calculated. However, as per the NEM model in (5) and the proposed HEM model in (8), they defined WAF(p) as a factor of the angle of incidence at the point of the intersection. Having obtained the initial points and terminal points of the points both S1 and S2, we calculate the magnitudes $\bar{V}1$ and $\bar{V}2$, using the Pythagorean theorem.
(24)$$\begin{array}{c} \bar{V}1_{ij}\;=\left(X_{2}\left(i\right)-X_{1}\left(l\right),\;\;Y_{2}\left(i\right)-Y_{1}\left(l\right)\right)\\ \bar{V}2_{pp}=\left(X_{4}\left(p\right)-X_{3}\left(p\right),\;\;Y_{4}\left(p\right)-Y_{3}\left(p\right)\right) \end{array}$$

Thus, applying Equation (24), we deploy the dot product to calculate the angle between the vectors as
(25)$$\begin{array}{c} Cos\varphi =\frac{\sum \left(\bar{V}1_{ij}\;.*\;\bar{V}2_{pp}\right)}{{\left\|\bar{V}1_{ij}\right\|}_{2}\;*\;\;{\left\|\bar{V}2_{pp}\right\|}_{2}\;}\\ \varphi =arccos\left(\frac{\sum \left(\bar{V}1_{ij}\;.*\;\bar{V}2_{pp}\right)}{{\left\|\bar{V}1_{ij}\right\|}_{2}\;*\;\;{\left\|\bar{V}2_{pp}\right\|}_{2}\;}\right) \end{array}$$

### 3.4. The RSSI Model Prediction Accuracy

RSSI prediction techniques at a particular RP within the dynamic environment vary a lot due to several factors of the AP location, height, physical characteristics of the environment, and the receiver’s device properties. At the same time, the time spent by specialized personnel to fingerprint the environment, in this case at 0.6 m sparsity, raises the cost and complexity of the fingerprint measurement technique; thus, an approach of prediction that reduces the construction cost and complexity of signal fingerprint databases will always be preferred. We base the prediction accuracy analysis on the minimization of errors between the prediction model values against the measurement fingerprint real values. In order to design a model that will reduce the cost, time, and expertise requirement for radio fingerprinting, which always increases with the dimensionality of the target space. That is, for corridor area 1 (57 m × 3.12 m) with 48 RPs, corridor area 2 (3.12 m × 16.96 m) with 20 RPs, lab area 3 (13.36 m × 8.9 m) with 20 RPs dimensionality sizes of the target space to fingerprint at the sparsity of 0.6 m, it will take the technical expert to spend 48 min, 20 min, 20 min for the 60 RSSI samples from respective AP’s, totaling to approximately 2 h, not including the time to shift the equipment from one RP to another. With an improved RSSI prediction model, the same process would take less than 15 s to run the algorithm on a computer, at the same time reducing the cost of the hardware to carry out the sampling fingerprinting.

Considering the evaluation of the proposed HEM prediction model in comparison to other prediction models against the measurement data samples, observation is made for the proposed (HEM) model in Equation (8) performing better than the NEM model as in Equation (3), MWM model as in Equation (2), and the OSM model as in Equation (1). Taking an example of AP1 as shown in Figure 5a, the proposed model attains a confidence probability of 10.98% with 0 dBm error prediction, relative to the 2.65% for the NEM model, 4.2% for the MWM model, 0% for the OSM model, respectively. In Figure 5b, we observe the similarity in the RSSI prediction of the NEM and the proposed, whereas in Figure 5c–f, the superiority of the HEM model is observed, followed by the MWM model with higher confidence probabilities as tabled in Table 4, Table 5, Table 6, Table 7, Table 8 and Table 9. The OSM model, like the initial model, does not take into consideration the multi-wall effect; the angle dependence effect, thus, falls short in prediction.

The proposed HEM’s RSSI prediction improves the accuracy of the RSSI estimation on the general floor target space than comparable models with an average prediction error (dBm) below 10 dBm for AP1, AP4, AP5, and below 20 dBm for AP2, AP3 than comparison models, as shown in Figure 6 and detailed in Table 10.

## 4. Localization Evaluation

System design for the online localization techniques can take the approaches based on the nearest neighbor classifiers and neural networks classifiers (Deterministic), whereas the other approaches are based on the Bayesian inference and statistical learning theory, respectively (Probabilistic). In this article paper, during the system design of our proposed system, we focused on the deterministic nearest neighbor classifier technique to analyze the performance of the models toward Wi-Fi indoor localization system; the nearest neighbor methods require a set of constant RSSI location fingerprints’ mean vectors or standard deviation as offline training set for the classification to carry out the UE’s estimate indoor locality.

### 4.1. The k-Value Effect

Simple in nature, the KNN algorithm, as a suitable approach for complex non-isotopic and varying environments, estimates the location from the first k nearest reference point neighbors resulting from the minimum Euclidean distance $d_{i}$ between Equation (13) test query RSSI values of the UE and the radio map RSSI, defined as
(26)$$\hat{\ell }\left({\bar{\vartheta }}_{j}\right)=\frac{1}{k}\sum_{i}^{k}\arg\min_{\ell_{i}}\left(d_{i}\right)$$

Determination of neighbor value k in KNN over the year has been investigated by which different researchers recommend retrials of different values until you obtain the right value for the specific research application. Complex indoor environments such as office buildings, shopping malls, university lecture halls, university libraries, or airport pedestrian spaces further complicated the determination of neighboring radio map reference points needed to be considered for a particular localization job.

We first evaluate the impact of the k-values towards minimization of the localization errors for the models in comparison with measurement samples at test reference points distributed on an indoor floor target space scaling from 0–50 m, a dimension relative to the size of the university lab floor. Our observed findings, as shown in Figure 7a, when *k* = 1, considering only one neighbor reference point in the KNN determination of the UE’s locality, within 10 m, the NEM model tends to attain a high confidence probability, whereas between 10 m to 20 m the proposed HEM model and the MWM models seem to perform similarly better than the NEM model and OSM model. In Figure 7b, when we increase *k* = 2, observations between 20 m to 30 m show the confidence gap of the MWM; the NEM models improved to 93% by 7% at 21 m and to 76% by 5%, respectively, while the proposed model seems to stabilize in a confidence range of 83% to 89%. In Figure 7c,d,f, increasing the *k*-neighbor to 3, 4, 5, and 6, respectively, gradually improves the confidence of the proposed model than comparison models; however, we further note that consideration of k = 5 and above does not bring forth to a significant change to indoor localization by the models.

In this article, after the above experimental analysis of the impact of k, rather than the experimental approach, we propose a simplified but composite solution to the empirical determination of the k neighbors for indoor positioning systems leveraging the dimensionality of the target space as in Table 11 the number of target spaces, and the number of reference points required within the space, taking note of the sparsity.

Having obtained the environmental parameters during the offline survey, we propose to calculate k using Equation (27).
(27)$$k=\frac{1}{P}\sum_{i}^{P}\left(\frac{Length\;\;(i)*width(i)}{SN\left(i\right)}\right)\;,\;\;\;\;\;\;\;\;\;i=1,\ldots ,3$$
where *P* is the total number of target spaces and *SN* is the offline total number of fingerprint RP’s. As per Equation (27), we determine k=4.

### 4.2. Localization towards Diverse Prediction Models

Secondly, evaluation of the localization systems accuracy has been set up using test sample RSSI predicted with the HEM model, the NEM model, the MWM model, and the measurement fingerprint, respectively. This approach is to account for the impact of indoor environmental propagation effects that impact the quality of RSSI received at a particular RP due to reflection, refraction, absorption of the radio signal, and diverse material properties that require extensive tuning during the offline stage. The KNN-based localization analysis Robustness of the HEM fingerprint using diverse predicted test samples by the NEM, MWM models indicate better average localization of 13% below 11 m than comparison fingerprints. As shown in Figure 8, we observe improved confidence in the proposed HEM model better than in the comparisons. In Figure 8a, on consideration of the HEM predicted test samples, we observe the HEM model initializing with 8% confidence at 0 Error (m) than 0% and 7% for the NEM and the MWM, respectively; further, outperformance is observed throughout the 9 m. Similar, outperformance is observed in Figure 8b with consideration of the NEM predicted test samples since they consider both the wall and angle of incidence effect in the model. However, on consideration with the MWM estimated test samples, we observe the localization by the MWM model outperforming the HEM and NEM though with lower confidence attained as shown in Figure 8c, for example, within 3 m as detailed in Table 12, Table 13 and Table 14, the MWM estimated test samples attains 21%, 21%, 23% confidence in relation to HEM’s 32%, 31%, 13%, NEM’s 26%, 21%, 18%, respectively. Consideration of the measurement test reference samples against the predicted fingerprint models, we observe the NEM initializing better than the rest until 10 m, with a 27% confidence; this could be attributed to the NEM’s break-point at 10 m from the transceiver, whereas the proposed HEM leverages known transceiver height and the signal wavelength in the determination of the breakpoint. Further, take note that indoor localization performance analysis tends to rely on the minimization of the distance error from the actual position of the UE at various heights. We have observed the proposed HEM model-built radio map attains better confidence probability with a significant gap than the comparisons between 10 m at 27% confidence to 24 m at 89 confidence; HEM model confidence between 24 m to 39 m seems stable while the NEM and the MWM model outperforms with better confidence. Analysis using the predicted samples and the measurement test samples, as shown in Figure 8 and Table 12, Table 13, Table 14 and Table 15, reveals a great diversity of the indoor environmental challenges towards factors that affect the signal propagation, and the accuracy performance improvement of the proposed HEM model using similar prediction parameters is observed. Dynamic tuning of the environmental factors, also discussed by recent research directions, will provide improvement in future prediction and indoor Wi-Fi localization systems.

### 4.3. The Fingerprint Localization Accuracy

Thirdly evaluation of the online localization systems accuracy has been set up using predicted fingerprints of the HEM model, the NEM model, the MWM model, and test fingerprints, respectively. Amongst the hearable APs, it is seen the proposed HEM fingerprint localization attains a confidence probability of 31% at 3 m, 55% at 6 m, and 78% at 9 m, outperforming the NEM with 26%, 43%, 62%, 62%, the MWM with 23%, 43%, 66%, respectively, as shown in Figure 9 and Table 16. The proposed HEM fingerprint, as per the analysis results, has demonstrated better performance, in turn reducing the cost of construction of the RSSI fingerprint than prior multiwall dependent models.

## 5. Conclusions

A novel RSSI prediction model that leverages the transceiver height and signals wavelength to fast construct an offline indoor Wi-Fi fingerprint of the target space is proposed. Fingerprinting has played a key part in signal space mapping tailored to the target space, which is once again leveraged during the online localization phase to minimize the distance error. We further proposed a method for the determination of *k* neighbors leveraging the target space dimensionality into the KNN algorithm towards Indoors fingerprinting. Experimental results carried out in the real indoor university lab floor environment show our proposed prediction model improves the prediction of the indoor signal strength of a mobile device by leveraging walls, signal angle of incidence, the height of the signal transceivers (APs), and the signal wavelength, as well as achieving a significant reduction of calibration time while providing a more comparable localization accuracy to that of the comparison prediction approaches as shown in Figure 9 by the KNN algorithm. We state that the proposed prediction algorithm has room for improvement in future research by enhancing the parameter estimation during the off-line stage and further developing dynamic intrusion sensing at a low cost.

## Figures and Tables

**Figure 1 sensors-22-09054-f001:**
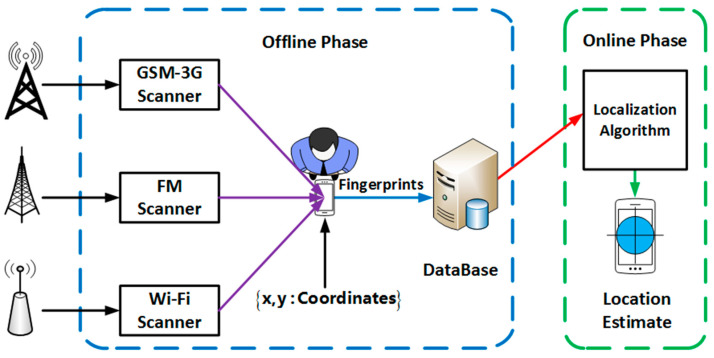
Fingerprint-based Localization System Architecture.

**Figure 2 sensors-22-09054-f002:**
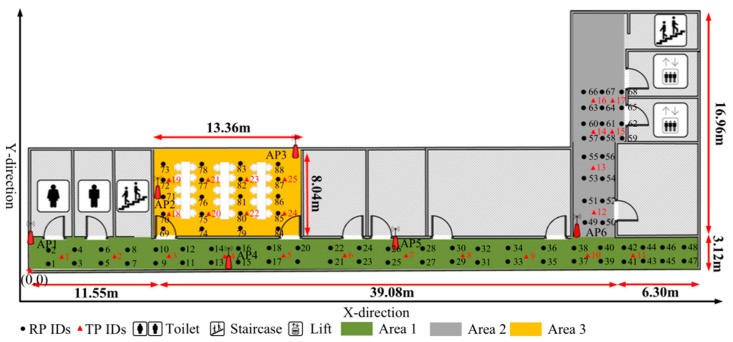
Test Bed. Corridor, office, lobbies environment scenarios.

**Figure 3 sensors-22-09054-f003:**
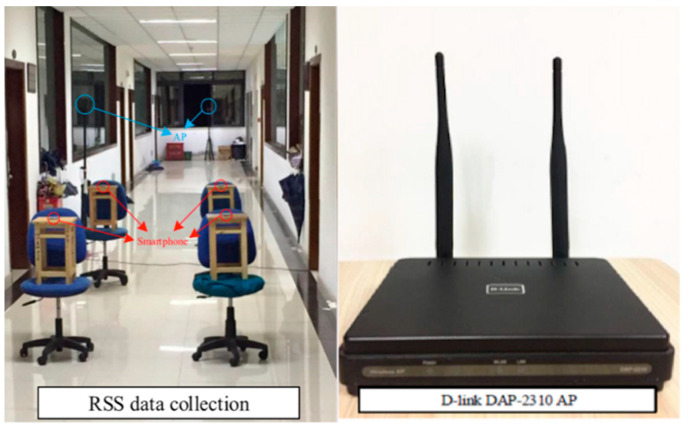
RSSI sampling.

**Figure 4 sensors-22-09054-f004:**
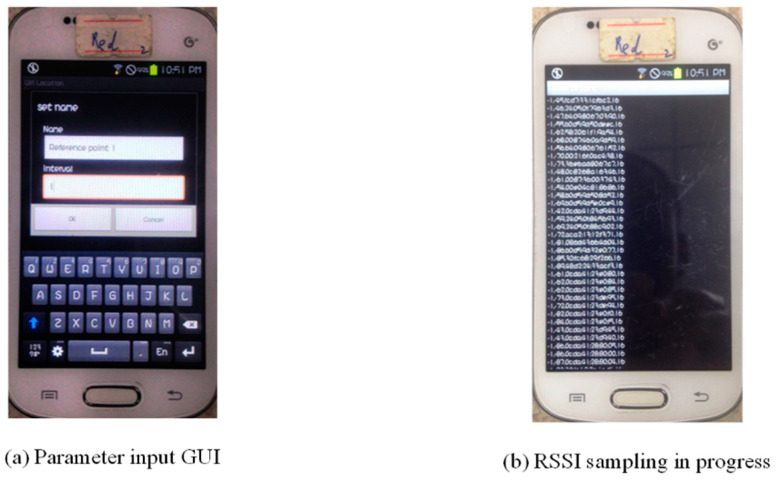
Data acquisition APP. (**a**) GUI interface for parameter input. (**b**) RSSI sampling at RP.

**Figure 5 sensors-22-09054-f005:**
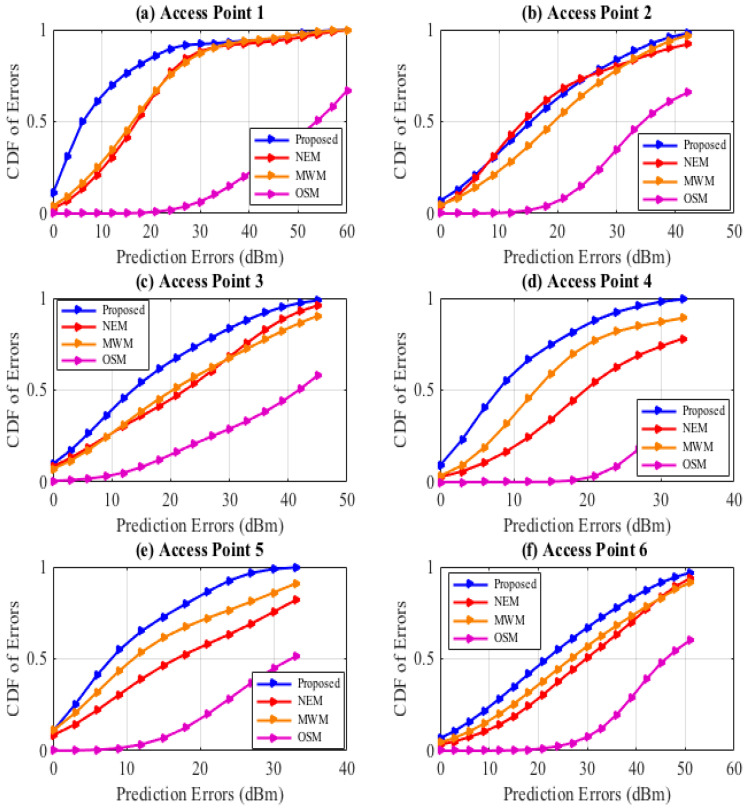
CDF Prediction Errors, AP1, AP2, AP3, AP4.

**Figure 6 sensors-22-09054-f006:**
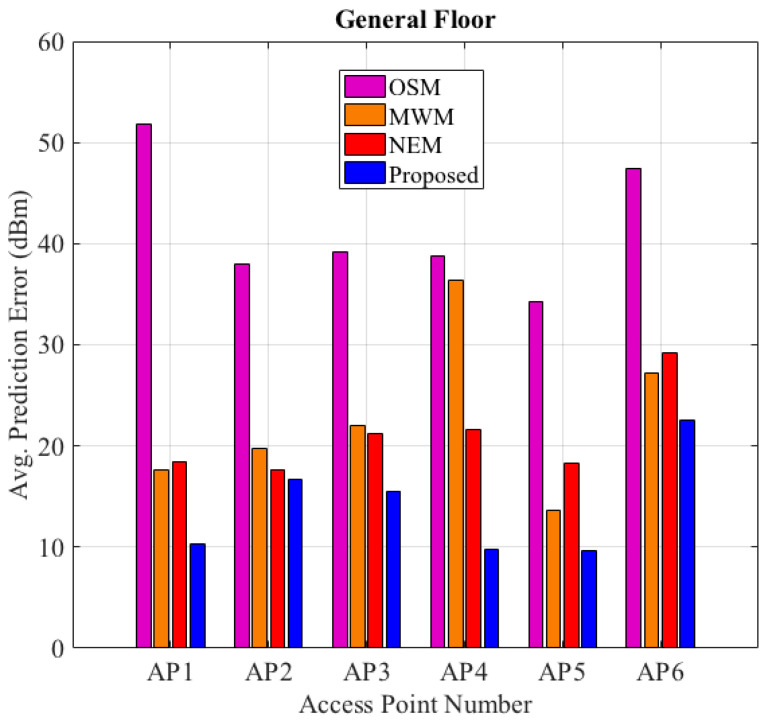
Prediction Errors. Respective AP’s average RSSI prediction comparison.

**Figure 7 sensors-22-09054-f007:**
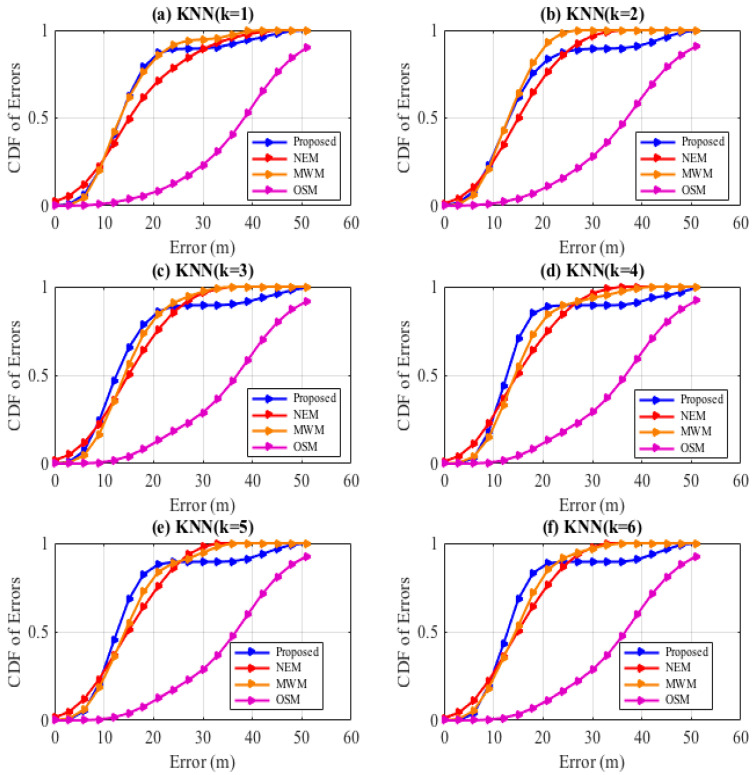
Localization. Comparison of the *k*-value with the KNN algorithm.

**Figure 8 sensors-22-09054-f008:**
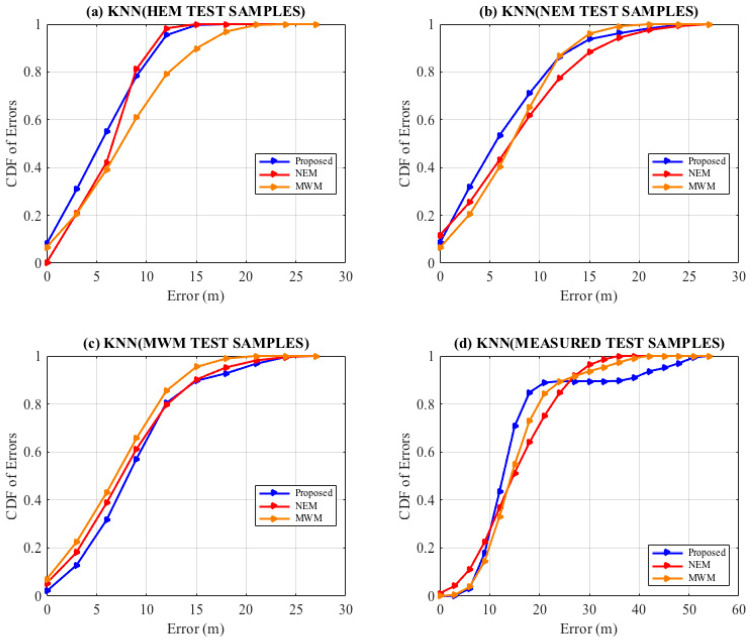
Localization. KNN Comparison by diverse predicted test samples.

**Figure 9 sensors-22-09054-f009:**
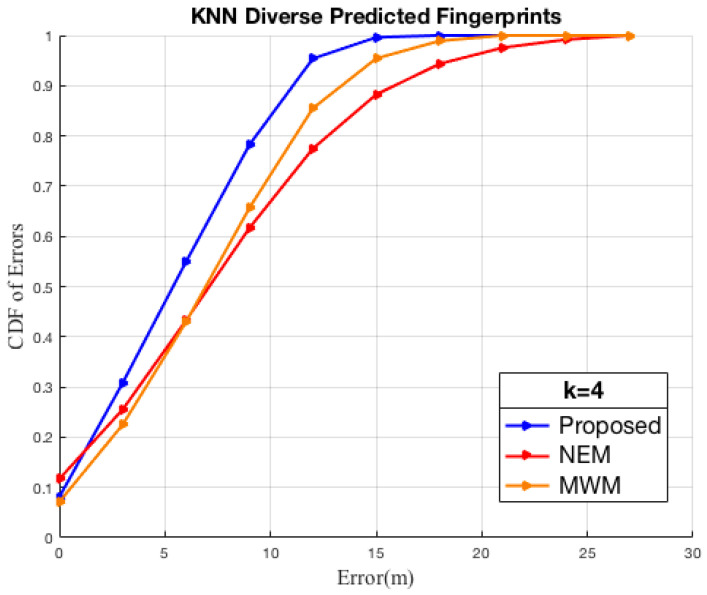
Localization. KNN Comparison by diverse predicted fingerprints.

**Table 1 sensors-22-09054-t001:** DAP-Link Aps, Scenarios, Coordinates, and Mac Address Matrix.

AP No	Scenario	Coordinates (m)	Mac Address
AP1	Corridor	(0.6, 9.96)	c0:a0:bb:27:88:20
AP2	Lab	(12.3, 16.02)	c0:a0:bb:29:d7:40
AP3	Lab	(24.6, 19.9)	c0:a0:bb:26:56:50
AP4	Corridor	(21.6, 8.7)	c0:a0:bb:27:90:88
AP5	Corridor	(31.14, 11.22)	c0:a0:bb:27:88:28
AP6	Corridor	(48, 11.82)	c0:a0:bb:26:d6:30

**Table 2 sensors-22-09054-t002:** Structure of the data log.

Interval	RSSI (dBm)	AP MAC Address	Time Stamp
−1	−42	c0:a0:bb:27:88:20	1
−1	−36	c0:a0:bb:29:d7:40	2
−1	−77	c0:a0:bb:26:56:50	3
…	…	…	…
−1	−88	c0:a0:bb:27:88:28	60

**Table 3 sensors-22-09054-t003:** Parameters.

	Values
*d_o_* (m)	1
WAF(*p*)	(4,4,10,4,4,4,4,4,15,15,15,15,4,4, 4,4,4,2.3,4,4,2.3,2.3,4,4,2.3,4,2.3, 4,4,2.3,2.3,4,2.3,4,4,4,4,2,2,4,2,2, 4,4,4,2,2,4,2,2,4)
*n*1	0.8
*n*2	2.5
*h* (m)	1.89

**Table 4 sensors-22-09054-t004:** AP1.

Errors (dBm)	0	12	24	36	48	60
Proposed	10.98%	69.74%	89.75%	93.27%	96.48%	99.97%
NEM	2.65%	30.5%	76.8%	91.7%	94.7%	99.7%
MWM	4.2%	34.4%	75.7%	92.5%	96.7%	99.8%
OSM	0.00%	0.00%	2.03%	14.83%	37.9%	66.9%

**Table 5 sensors-22-09054-t005:** AP2.

Errors (dBm)	0	12	24	36	42
Proposed	6.97%	39.54%	72.26%	92.62%	98%
NEM	3.93%	42.65%	73.24%	86.93%	91.99%
MWM	4.48%	28.25%	63.84%	89.39%	96.91%
OSM	0.0%	0.65%	14.81%	54.36%	65.76%

**Table 6 sensors-22-09054-t006:** AP3.

Errors (dBm)	0	12	24	36	45
Proposed	9.88%	45.57%	73.28%	92.05%	98.79%
NEM	8.4%	30.34%	53.35%	82.58%	96.01%
MWM	7.14%	31.33%	56.99%	77.42%	90.44%
OSM	0.5%	5.07%	20.72%	38.12%	57.97%

**Table 7 sensors-22-09054-t007:** AP4.

Errors (dBm)	0	6	12	18	24	33
Proposed	9.11%	40.28%	66.47%	81.54%	92.45%	99.4%
NEM	2.71%	10.45%	24.29%	44.18%	62.65%	78.06%
MWM	3.28%	18.86%	45.40%	69.48%	81.88%	89.2%
OSM	0.0%	0.0%	0.0%	0.8%	8.64%	41.92%

**Table 8 sensors-22-09054-t008:** AP5.

Errors (dBm)	0	6	12	18	24	33
Proposed	10.8%	41.21%	64.96%	79.72%	92.59%	99.68%
NEM	8.3%	22.01%	38.91%	52.55%	63.35%	81.98%
MWM	11.0%	31.91%	53.54%	67.31%	76.54%	90.93%
OSM	0.0%	0.5%	3.37%	12.44%	28.35%	51.35%

**Table 9 sensors-22-09054-t009:** AP6.

Errors (dBm)	0	12	24	36	48	51
Proposed	6.78%	28.15%	54.91%	77.87%	94.5%	96.83%
NEM	3.26%	14.05%	37.25%	63.25%	89.33%	93.81%
MWM	4.46%	19.82%	44.41%	68.07%	87.61%	91.54%
OSM	0.0%	0.0%	2.44%	19.63%	54.52%	60.07%

**Table 10 sensors-22-09054-t010:** Average prediction errors evaluation.

Access Point	AP1	AP2	AP3	AP4	AP5	AP6
Proposed	10.36094	16.65165	15.5347	9.716051	9.685779	22.52913
NEM	18.47965	17.58727	21.23628	21.64572	18.24757	29.19121
MWM	17.58734	19.7279	22.03516	15.28833	13.67405	27.2322
OSM	51.85314	38.0159	39.1757	38.7193	34.32284	47.48057

**Table 11 sensors-22-09054-t011:** Environment parameters.

	Area 1	Area 2	Area 3
Length *(i)*	56.93 m	16.96 m	13.36 m
Width *(i)*	3.12 m	3.12 m	8.4 m
SN *(i)*	48	21	21

**Table 12 sensors-22-09054-t012:** KNN with HEM Simulated Test Samples.

Error (m)	0 (m)	3 (m)	6 (m)	9 (m)	12 (m)	15 (m)	18 (m)	21 (m)
Proposed	8%	31%	55%	78%	95%	100%	100%	100%
NEM	0%	21%	42%	81%	98%	100%	100%	100%
MWM	7%	21%	39%	61%	79%	90%	97%	100%

**Table 13 sensors-22-09054-t013:** KNN with NEM Simulated Test Samples.

Error (m)	0 (m)	3 (m)	6 (m)	9 (m)	12 (m)	15 (m)	18 (m)	21 (m)
Proposed	8%	32%	53%	71%	86%	94%	96%	98%
NEM	12%	26%	43%	62%	77%	88%	94%	98%
MWM	6%	21%	40%	65%	87%	96%	99%	100%

**Table 14 sensors-22-09054-t014:** KNN with MWM Simulated Test Samples.

Error (m)	0 (m)	3 (m)	6 (m)	9 (m)	12 (m)	15 (m)	18 (m)	21 (m)
Proposed	2%	13%	32%	57%	80%	90%	93%	97%
NEM	5%	18%	39%	61%	80%	90%	95%	98%
MWM	7%	23%	43%	66%	85%	95%	99%	100%

**Table 15 sensors-22-09054-t015:** KNN with Measurement Test Samples.

Error (m)	0 (m)	3 (m)	6 (m)	9 (m)	12 (m)	15 (m)	18 (m)	21 (m)
Proposed	0%	0%	3%	18%	44%	71%	85%	89%
NEM	1%	4%	11%	22%	37%	51%	64%	75%
MWM	0%	1%	4%	15%	33%	55%	73%	84%

**Table 16 sensors-22-09054-t016:** KNN with diverse fingerprints.

Error (m)	0 (m)	3 (m)	6 (m)	9 (m)	12 (m)	15 (m)	18 (m)	21 (m)
Proposed	8%	31%	55%	78%	95%	100%	100%	100%
NEM	12%	26%	43%	62%	77%	88%	94%	98%
MWM	7%	23%	43%	66%	85%	95%	99%	100%

## Data Availability

Data underlying the findings of the study is available on request.

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
