# Peer review of "RSSI Fingerprint Height Based Empirical Model Prediction for Smart Indoor Localization"

_sensors, 2022, doi:10.3390/s22239054_

Round 1

Reviewer 1 Report

The manuscript "RSSI Fingerprint Height Based Empirical Model Prediction for Smart Indoor Localization" addresses a topic of interest to a broad audience and fits the journal's scope. This article proposed a technique based on a novel RSSI prediction method that improves the estimation and localization accuracy for complex indoor environments at a low cost of fingerprinting since no additional hardware is required; from the experimental results, the proposed method achieved improvement in stability and precision.

1.     Page 1, line 13 RSSI should be Receive Signal Strength Indicator.

2.     Page 8, line 324-328 " There are 25 test points, that is, in Figure 2, the black circle, the red triangle represents the reference points (RP's), and the test points (TP's) respectively." à There are 25 test points (TP's), that is, in Figure 2, the black circle, the red triangle represents the reference points (RP's), respectively.

3.     When you want to use the English abbreviation, when it first appears, write it all and then use the acronym (AP, MD, LOS, RP…).

      4. Could you give readers a more confident k value from your findings? Or tell readers when uses which k value. It is a contribution of this article.

Author Response

Dear Reviewers.

Greetings!

We are thankful for  the careful review's and great observation, we have carried the review and we hereby resubmit our manuscript “RSSI Fingerprint Height based Empirical Model Prediction for Smart Indoor Localization” That presents a novel RSSI prediction approach leveraging the angle of incidence, wall factors, transceiver height and signal wave length. We are optimistic with the editorial team and may not hesitate to contact us if changes of details and review are required.

Regards,

Arigye (on behalf of authors)

Wilford Arigye

Chongqing Key Lab of Mobile Communications Technology

Institute of Wireless Location and Space Measure

Chongqing University of Posts and Telecommunications

Chongqing 400065, China

E-mail: L201910017@stu.cqupt.edu.cn

Reviewer 2 Report

This paper experimentally evaluates and proposes novel techniques for RSSI distance prediction, leveraging transceiver height, and Fresnel ranging in a complex indoor environment to better suit the path loss of RSSI at a particular RP and time, which further contributes greatly to indoor localization. It seems interesting and has good work, but there are some comments and suggestions as follows:

- Abstract: several short terms such as RP,AP1, NEM, MWM, HEM are not defined. The authors should mention the percentage localization accuracy compared to other algorithms. 

- Please either use capitalize first letter work or lowercase when you define the short terms. For example, Global Positioning System (GPS), location-based services (LBS)- NLOS and AP are not defined. - I suggest adding the paper's organization in the last paragraph of section 1. 

- Section 2, some short terms such as MD, LOS, RP, RTLSs are not defined.

- On page 6, line 250 Where P and Q are the number of walls and floors between the transmitter and receiver, respectively. P and Q should be small letters based on equation 2. 

- There is no effect on FAF in equations 2 and 4 since you're not considering multi-floor budling.

- In section 3, how do you measure the x,y coordinates of each RP during the offline phase?

- The authors mentioned "Considering the Evaluation of the proposed prediction model" on page 12, line 414, but I did not see any proposed model. 

- How did you calculate the confidence probability? 

- Figure 5, How you calculate the parameters of equations 2,3 and 8, such as d,d0, WAF, FAF, cosθ. 

- In section 4, the reviewer is interested in seeing the localization accuracy of each area and also overall localization accuracy.  Figures 7 and 8 show a high localization error for all models including the "proposed model". It can be observed that the average localization error of the proposed model is more than 20 meters which are not acceptable. There are many works that have achieved an average localization error below 1 meter. Please see some works such as: A three-dimensional pattern recognition localization system based on a Bayesian graphical model; Improving accuracy in indoor localization system using fingerprinting technique.

- The authors should present their results in average localization error and overall system accuracy.

Author Response

Dear Reviewers.

Greetings!

We are thankful for  the careful review's and great observation, we have carried the review and we hereby resubmit our manuscript “RSSI Fingerprint Height based Empirical Model Prediction for Smart Indoor Localization” That presents a novel RSSI prediction approach leveraging the angle of incidence, wall factors, transceiver height and signal wave length. We are optimistic with the editorial team and may not hesitate to contact us if changes of details and review are required.

Regards,

Arigye (on behalf of authors)

Wilford Arigye

Chongqing Key Lab of Mobile Communications Technology

Institute of Wireless Location and Space Measure

Chongqing University of Posts and Telecommunications

Chongqing 400065, China

Reviewer 3 Report

In this paper, the authors presents a model that combines general indoor pathloss model and the factors of wall attenuation (with angle factor) and access point height.

The experimental results show that the proposed method has better accuracy on the prediction of RSSI.

The major concerns are as follows:

1. The height of APs has been investigated by a few research groups for indoor localization, not as claimed "not addressed leveraging the height". Please perform a more comprehensive literature research and include related publications, especially in last decade.

2. In the proposed model, there is only one factor "AP transceiver height" introduced, is this the height of APs against level ground, sea level, or any other reference? For example in Okumura Hata model, both the factors of base station height and user equipment height are against level ground. Please explain.

3. The RSSI prediction result looks good, yet when applying KNN method, the distribution of localization error seems diverged quite a lot. There are roughly 12% of cases where the localization error is around 40m or higher. The root cause should be analyzed and explained.

Author Response

(The authors gave the same response as above.)

Round 2

Reviewer 2 Report

Thanks for addressing my comments, but still few minor corrections as follows:

 - The authors should eliminate FAF in Equation 8 since the proposed model is not considered a multi-floor building. 

- Section 2 lacks most related works such as A three-dimensional pattern recognition localization system based on a Bayesian graphical model; Robust 3D indoor positioning system based on radio map using Bayesian network;  Improving accuracy in indoor localization system using fingerprinting technique. You should include them.

- The localization error with a minimum average of 6 meters is still very high. Thus, you should justify your work. 

Author Response

Greetings Reviwer 2, 

We thanks you for the careful review of our manuscript, and we herby attach the response for further consideration. 

Regards!

Reviewer 3 Report

Concerns 1 & 2 are adequately addressed, but concern 3 was not responded in a proper manner. The response to concern 3 did not explain why "in roughly 11% of the tests, the localization error of proposed approach is higher than 30m"

Please add your discussion in the manuscript.

Author Response

Dear Reviewer, 

Greetings, we thank you for the careful review of our manuscript and we have carried our a review for your ponderation. Hope for the best. 

On hqebalf of the Authors 

Regards!
